

# Bayesian Network Model for Flood Forecasting Based on Atmospheric Ensemble Forecasts

Leila Goodarzi[1], Mohammad Ebrahim Banihabib[1], Abbas Roozbahani[1] and Jörg Dietrich[2]

5   1 Dept. of Irrigation and Drainage, College of Aburaihan, University of Tehran, Tehran, Iran
Institute of Hydrology and Water Resources Management, Leibniz Universität Hannover, Hannover, Germany

Correspondence to: Jörg Dietrich (dietrich@iww.uni-hannover.de)

**Abstract.** The purpose of this study is to propose the Bayesian Network (BN) model to estimate flood peak from Atmospheric Ensemble Forecasts (AEFs). The Weather Research and Forecasting model was used to simulate historic storms using five cumulus parameterization schemes. The BN model was trained to forecast flood peak from AEFs. Mean Absolute Relative Error was calculated as 0.076 for validation data while it was calculated as 0.39 in artificial neural network (ANN) as a widely used model. It seems that BN is less sensitive to small data set, thus it is more suited for
forecasting flood peak than ANN.

**Keywords:** Artificial neural networks; Bayesian networks; ensemble flood forecasting; WRF model.

## 1 Introduction

Floods are the most threatening natural disaster across the world (Hénonin et al., 2010). Studies show that over 80% of the
cities of Iran are at the risk of flooding (Chitsaz and Banihabib, 2015). Flood warning is an efficient way to reduce the flood damage. However, many flood forecasting systems in the world rely on observed rainfall and thus, the lead time of these systems is often short for small basins (Banihabib and Arabi, 2016). Numerical Weather Prediction (NWP) models can be used to increase the lead time of flood warning by using in advance forecasts of rainfalls. Although the combination of NWP and hydrological models can significantly increase the flood warning lead-time rather than using observed rainfalls, the
deterministic weather prediction doesn't reflect the existing uncertainties. Thus, in the last decades, many operative and research on the flood forecasting systems around the world are increasingly employing ensembles of NWPs instead of single deterministic forecasts which have considerable uncertainties. Ensemble methods are considered to be an effective way to estimate the probability of future states of the atmosphere by addressing uncertainties present in initial conditions and in model approximations (Tennant, et al, 2007). Various approaches have been developed to produce atmospheric ensemble





forecasts including perturbing the initial conditions, perturbing the input parameters of the model, using multi-model ensembles and using different parameterization schemes (Yang et al., 2012).

One of the most important parameterization schemes is the cumulus parameterization. NWP models often use Cumulus Parameterization Schemes (CPS) to consider the effects of cumulus clouds which are not represented in the modelling as
they are much smaller than the model grid size (Pennelly et al., 2014). Common CPS are presented in table 1.

Kerkhoven et al. (2006) compared various CPS for a summer monsoon in east China and found that the Kain–Fritsch scheme is the best scheme at simulating moderate rainfall depths. Pennelly et al. (2014) applied the WRF model with diverse cumulus parameterization schemes for three flood events in Alberta, Canada, and they showed that the Kain–Fritsch and explicit cumulus parameterization schemes were the most accurate for simulating the rainfalls. Other studies indicated that
ensemble forecasting is promising for predicting heavy rainfall ((Deb et al., 2008); (El Afandi et al., 2013); (Li et al., 2014)). Ensemble meteorological forecasting is widely coupled with a hydrological model to predict stream flow ensembles. Li et al. (2017) coupled the WRF model with a distributed hydrological model for flood forecasting in a large watershed in southern China. The results suggest that the simulated floods are rational and could benefit the flood management communities due to its longer lead time. Rogelis and Werner (2018) assessed the potential of NWP models for flood early warning in tropical
mountainous watersheds. The results showed that the streamflow forecasts resulted from a hydrological model forced by post-processed rainfall using the WRF added value to the flood early warning systems.

Only few case studies report how flood hydrographs derived from Atmospheric Ensemble Forecasts (AEFs) can be converted into warning decisions during a flood event , E.g. using the exceeding ensemble members (Li et al. 2017; Abebe and Price (2005)), using the quantile of the predicted flow ensemble (Dietrich et al. (2009b)), integrating ensemble rainfall
forecasts, rainfall thresholds and a real time data assimilation method (Yang et al. (2016)) and and combining high capacity cloud computing and an interactive web service in a social network (Shi et al. (2015)).

Other concepts of deriving a single (deterministic type) warning indicator from ensembles are weighting of ensemble members e.g. averaging by Bayesian model (Raftery et al., 2005) or by machine learning (Doycheva et al., 2016) or by reduction of members to create a multi-model super-or sub-ensemble (Dietrich et al., 2009a).
According to previous studies, converting the ensemble forecasts into warnings and also deriving a single warning indicator from ensembles are not yet adequately considered and remains a challenging question in ensemble based flood warning. The main objective of this study is to propose the BN model to estimate the flood peak from a meteorological ensemble forecast without employing a hydrological model. BN has been widely used by researchers in many water resources fields. Applications of BN in water resources can be found in Mediero et al. (2007), Vogel et al. (2012); Sharma and Goyal (2016)
and Shin et al. (2016). Phan et al. (2016) reviewed 111 BN applications in water resources management but only 4 were in the domain of river flow, 5 were in operational decision making context and none in operational flood warning. BN application in ensemble flood forecasting has not been reported yet to our best knowledge.

In previous studies, meteorological ensemble forecasts are coupled with a hydrological model to predict a set of flood hydrographs with different peak discharge. Ensemble decision making according to a range of possible flood peaks is a





challenging issue especially in case of equal likelihood of each ensemble member. In the present study, the hydrological model is replaced by a Bayesian network for deriving a single warning indicator from atmospheric ensemble forecasts.

The purpose of the present study is therefore to predict the flood peak addressing the uncertainties and the probability of occurrence of each ensemble member. As a case study, flood peaks were forecasted in a relatively small mountainous basin,

5 Kan Basin, Tehran, Iran. The Weather Research and Forecasting (WRF) model was used to simulate 14 historic precipitation events using five different cumulus parameterization schemes. Then atmospheric ensemble forecasts were coupled to the BN to estimate the flood magnitude for an ensemble forecasting, from which flood warnings could be derived. Forecasting performance of the BN was compared with the results obtained from an artificial neural network (ANN) as a widely used models.

## 10 2 Data and methodology

### 2.1. Study Area

The case study of this research is Kan Basin, Tehran, Iran with an area of 197 km². The geographical limits lie between 35°46′ N to 35°58′ N latitudes and 51°10′ E to 51°23′ E longitudes. Figure 1 shows the location of the study area. Average elevation is 2428.7 m above sea level and the annual rainfall is about 600 mm. The rainfall data was from Emamzadeh-

15 Davood rainfall station and the flow data was collected from Sooleghan hydrometric station that is located downstream of the basin as shown in Figure 1. The time of concentration (Tc) of the basin is about 3 hours, so the NWP models can significantly increase the lead time of flood warning compared to using observed precipitation. Since the increasing of lead time decreases the accuracy of NWP forecasts (Sikder and Hossain, 2016), thus the forecasting was conducted one day before the observed event. A flow chart of the proposed flood forecast approach is presented in Figure 2 and the precipitation

20 and streamflow data are presented in table 2.

### 2.2. The Weather Research and Forecasting model (WRF)

The Weather Research and Forecasting (WRF) model was used to simulate 14 historic heavy precipitation events that caused floods in the study area. In this study, WRF version 3.8 was employed with 3 domains and one hour temporal resolution. Horizontal resolution of domains are 45 km, 15 km and 5 km, respectively. Figure 3 shows the WRF domain setup using an

25 inter-active nested domain inside the parent domain. The outer (the coarsest) domain covers Iran, the middle domain covers the northern part of Iran and the inner domain covers the study area and only the meteorological information from this domain was used for forecasting of flood in the study basin.

The 6-hour analyses of the Global Forecast System (GFS), produced by the National Centers for Environmental Prediction (NCEP), were used as the initial conditions of the WRF. The model settings were based on the Noah land surface model

30 (Chen and Dudhia, 2001), the Rapid Radiative Transfer Model (RRTM) longwave radiation scheme (Mlawer et al., 1997),





the Dudhia shortwave radiation model (Dudhia, 1989), the Yonsei University (YSU) planetary boundary layer scheme (Hong et al., 2006) and the WRF Single-Moment (WSM) 3-class microphysics scheme (Hong et al., 2004). Because of the importance of cumulus parameterization in hydrological purpose, an ensemble was created by using five cumulus schemes including KF, BMJ, GR3D, MSKF and GDE cumulus scheme. The atmospheric ensemble forecasts were fed into the

Bayesian Network to estimate flood peak flow.

### 2.3. Bayesian Network

This study proposed a probabilistic model to make the flood forecasts and to estimate the flood magnitude based on Bayesian networks (BN) for an ensemble forecasting. BNs are a class of probabilistic graphical models composed by a set of random variables and directed acyclic graphs (DAG) to show the potential dependence between variables (Scutari, 2017).

The node at the start of an arrow is casual or preceding event that is called parent node and the node at the head is an outcome event that is called child node. Each node is labelled with a conditional probability table (CPT) based on prior information or statistically observed correlations that shows the strengths of the influences of the parent nodes on the child node. In general, assuming random variables with domain size d, conditional probability table of a child node with n parents needs one to specify dn+1 probabilities (Li et al, 2011).

The goal is to calculate the posterior conditional probability distribution of each of the possible unobserved causes given the observed evidence, i.e. P [Cause ∣ Evidence].

However, in practice we are often able to obtain only the converse conditional probability distribution of observing evidence given the cause, P [Evidence ∣ Cause]. The whole concept of Bayesian networks is built on Bayes theorem, which helps us to express the conditional probability distribution of cause given the observed evidence using the converse conditional

probability of observing evidence given the cause as Eq (1):

$$P [\text{Cause} \mid \text{Evidence}] = P[\text{Evidence} \mid \text{Cause}] \frac{P [\text{Cause}]}{P[\text{Evidence}]} \tag{1}$$

Any node in a Bayesian network is always conditionally independent of its all non-descendants given that node's parents. The conditional probabilities are represented in the form of Conditional Probability Distribution (CPD) if the nodes represent a continuous variable or Conditional Probability Table (CPT) if the nodes represent a discrete variable. The joint probability

(Pb) can be defined as the product of the local conditional distributions as given in Eq (2):

$$P_b(x_1. x_2. \dots. x_n) = \prod_{i=1}^{n} P_b(x_i \mid x_{i+1}. \dots. x_n) \tag{2}$$

In a BN, a node xi is independent of all other nodes except its parents (Sharma and Goyal, 2016). A simple example of BN is presented in Figure 4. The joint probability for this simple network can be defined as Eq (3):


$$p(a.b.c) = p(a) \times p(b \mid a) \\ \times p(c \mid a.b)$$

(3)

The graph containing nodes and arrows is called BN structure (BS). Learning a Bayesian Network includes two aspects: structure learning and parameter learning.

Structure Learning: The purpose of structure learning is to determine the best structure, which maximizes the conditional probability P(BS|D), where BS is the BN structure and D is the given data (Sharma and Goyal, 2016).Structure learning consists in finding the DAG that encodes the conditional independencies present in the data. This has been achieved in the literature with constraint-based, score-based and hybrid algorithms (Scutari, 2017). Some common structure learning techniques are K2 algorithm (Cooper and Herskovits, 1992, Amirkhani and Rahmati, 2015) and MCMC algorithm (Madigan

et al., 1995). However, BS can be easily defined if the relationship between child nodes and parent nodes is known. In the present study, the flood is influenced by atmospheric ensemble forecasts, base flow of the river and antecedent rainfall, so the BS is known.

*Parameter Learning:* Bayesian network conditional probability tables (CPTs) can be learned when the BN structure is known. Different parameter learning algorithms have been presented, including expectation maximization, Markov Chain

Monte Carlo methods such as Gibbs sampling, and gradient descent methods (Reed and Mengshoel, 2014). In this study, Expectation Maximization (EM) was used for Bayesian Network parameter learning. The EM algorithm is an iterative method that performs a number of iterations, each of which calculates the logarithm of the probability of the data given the current joint probability distribution. This quantity is known as the log-likelihood, and the algorithm attempts to maximize likelihood estimators (Bergmann and Kopp, 2009). In the Hugin Tool convergence is achieved when the difference between

the log-likelihoods of two consecutive iterations is less than or equal to the numerical value of a log-likelihood threshold times the log-likelihood. Alternatively, the user can specify an upper limit on the number of iterations to ensure that the procedure terminates.

Our proposed ensemble forecasting using a BN model has the following four steps:

1) Selecting relevant variables and spatial units,

2) Creating training data set for the model,

3) Learning the model using HUGIN software version 8.3 (further developed from original work of Lauritzen and Spiegelhalter, 1988) and

4) Evaluating the performance and accuracy of the model.

In the present study, the flood peak is the response variable that is influenced by some predictor variables including

ensemble rainfall forecasts, base flow of the river and antecedent soil moisture. Base flow of the river is the normal day to day discharge. Antecedent recharge flow was used as the base flow of the river. The catchment's antecedent soil moisture represents the relative wetness prior to the flood event and can have an important influence on flood response. Because of



the lack of soil moisture data in the Kan basin, antecedent rainfall was used to represent the soil moisture. Antecedent rainfall is the total precipitation amount that occurred in the 24 hours before the start of the event.  This study was performed on 14 historical storms. It should be noted that approx. 70% of the available data (10 storm events) is allocated for training and the remaining (4 storm events) data are used for validation. The data sample is relatively small due to the following

reasons:

1) NCEP (GFS - FNL) data are not available for some historical storms.

2) During the above-mentioned period, a small number of actual flood events occurred in the study area, since the basin is located in a semi-arid region.

Considering the relatively small sample size, we proposed using the BN that is less sensitive to small data set size in

comparison with ANN. Some advantages of BN are:

**Suitable for small and incomplete data sets:**

A very useful advantage of BN is that there are no minimum sample data sizes needed to perform the analysis, and BN take into account the complete data set (Myllymaki et al., 2002). Also, Kontkanen et al. (1997) demonstrate that BN can show good accuracy of prediction even with rather small data set. Furthermore, Zhang and Bivens (2007) showed that BN is less

sensitive to small data set size in comparison with ANN.

**Structural learning possible:**

It is possible to use data and also subject matter knowledge to learn the structure of BN. This is an aspect of active research, and though the statistical theory is well understood, the techniques are still under development (Jensen, 2001).

**Fast responses:**

Since BN is analytically solved, it can provide fast responses to requests once the model is compiled. The compiled form of a BN comprises a conditional probability distribution for each combination of variable values, and thus can provide any distribution instantly, in contrast to the other simulation models in which the results need to be simulated, which can take very long (Uusitalo, 2007). Thus, BN are recommended for operational ensemble forecasting in particular in fast reacting basins, where a high number of forecasts must be simulated within a short time.

**2.4. Artificial Neural Networks (ANN)**

Artificial Neural Networks (ANN) are used as an alternative of statistical models in different aspects including clustering analysis, estimation, sample recognition etc. (Mammadov et al, 2005). An ANN model is basically an engineering method of biological neurons. It is constructed by input, output and hidden layers. ANN consist of a large number of simple processing elements, which are interconnected with each other and layered also (Sharma et al, 2012).

Typically, there are four distinct steps in developing an ANN model. The first step is data transformation or scaling. The input and output variables are first normalized linearly in the range of 0 and 1 using the following equation:

$$\bar{X} = \frac{X - X_{min}}{X_{max} - x_{min}} \tag{4}$$





Where $\bar{X}$ the normalized value of the X. Xmin and Xmax is are the minimum and maximum of data, respectively. The main purpose for standardizing the data is that the variables are usually measured in different units. By normalizing the variables in dimensionless units, the arbitrary effect of similarity between objects is removed  (Aichouri et al., 2015).

The second step is the network architecture definition in which the number of hidden layers, the number of neurons in each

layer, and the connectivity between the neurons are determined. The number of neurons and hidden layers is problem dependent and is estimated by the trial and error technique or expert experience. A synaptic weight is allocated to each link to represent the relative connectivity strength of two nodes at both ends in predicting the input-output relationship (Raju et al, 2011). A typical ANN architecture is presented in Figure 5.  In this study, the output from the model is the flood peak and the input variables are atmospheric ensemble forecasts, base flow of the river and antecedent rainfall. The third step is using

a learning algorithm to train the network to predict correctly to the set of inputs. There are several learning algorithms. In the present study, the most widely used feed forward error back propagation algorithm was used for training because of the good performance of this algorithm in the previous studies (Raju et al, 2011; Banihabib et al, 2015, ASCE, 2000; Sarkar and Kumar, 2012). The success of an ANN application depends on the quality and also the quantity of the available data (Cheng et al., 2017). Final step is the validation, in which the performance of the trained ANN model is evaluated using statistical

criteria (Sarkar and Kumar, 2012).

### 2.5 Statistical criteria for validation

 In the present study, Mean Absolute Relative Error ($MARE$), Mean Relative Bias Error ($MRBE$) and regression coefficient ($r$) were used for performance evaluation of the model as given in the following equations:

$$MARE = \frac{1}{n}\sum\frac{|O_i - F_i|}{O_i} \tag{6}$$

$$MRBE = \frac{1}{n}\sum\frac{O_i - F_i}{O_i} \tag{7}$$

$$r = \frac{n(\sum OF) - (\sum O)(\sum F)}{\sqrt{[n\sum O^2 - (\sum O)^2][n\sum F^2 - (\sum F)^2]}} \tag{8}$$

$O_i$ is the observed value, $F_i$ is the predicted value and $n$ is the total number of data sets.

### 3. Results and discussion

### 3.1. Rainfall verification using the WRF model

In this section, the comparison between the observed and predicted precipitation obtained from the WRF model is addressed. As mentioned earlier, the WRF model was used to simulate 14 historic precipitation events and the results for some events are presented here. Figures 6 illustrates the predicted cumulative rainfall and the observed cumulative rainfall for these events. In general, the results show that the WRF model was able to capture the heavy rainfall events. The uncertainties in



the predicted rainfall lead to a large spread of the ensemble members and this is why the uncertainty in rainfall forecasting becomes important.

The ensemble precipitation illustrate that both overestimation and underestimation of precipitation occurs using various schemes. Overestimation is very noticeable for the early hours of forecasting while for last period of the event, underestimation occurs in some schemes.

From the case study, the results of precipitation forecast using different cumulus schemes by the WRF model can be significantly different. Therefore, it is necessary to forecast precipitation by implementing various physics schemes, especially different microphysical schemes. Furthermore, it can be inferred that the difference between observed and predicted rainfall is mainly caused by the initial condition in the NWP models, thus the atmospheric ensemble forecasts can be produced also by perturbing the initial conditions.

### 3.2. Bayesian Network Verification

The atmospheric ensemble forecasts were fed into the BN to estimate flood peak flow. Ten various models were developed using various combinations of predictors. In all of the combinations, flood-peak discharge is the predicting variable. Table 3 shows the accuracy of the model for different combinations of predictors to compare the performance of the prediction. The performance of the model was evaluated by MARE and R2. It is clear from Table 3 that maximum hourly rainfall outperformed accumulated rainfall as predictor variable (No. 2 in Table 4). It shows for the relatively short concentration-time basin, Kan basin, that cumulative precipitation is not a good indicator to predict the flood peak and maximum hourly rainfall provides better results. Thus maximum hourly rainfall was used in combinations of other predictor variables. This can also be seen by comparing combination No. 5 and 9 that there is no considerable decrease in accuracy by deleting the Multi-scale Kain-Fritsch scheme, consequently it can be concluded that MSKF is the least accurate cumulus scheme. It was also found that by deleting the Kain-Fritsch scheme in combination (No. 6 in Table 3) the accuracy is significantly decreased. Thus, the Kain-Fritsch is the most efficient cumulus parameterization scheme in the study area. Other studies on precipitation prediction have also shown similar results. Pennelly et al. (2014) showed that the Kain-Fritsch cumulus parameterization schemes is the most accurate in simulating heavy precipitation across three summer events. Liang et al. (2004) showed that the Kain-Fritsch scheme works better in the Southeast of United States where convection is largely governed by the near-surface forcing.

According to Table 3, the best results were obtained for combination No. 5. The proposed structure of this combination is composed of eight nodes as shown in Figure 7. This graphical structure defines the cause-effect relationships among the variables. Atmospheric ensemble forecasts, base flow of the river and antecedent rainfall are the parent nodes and flood peak is the child node. It can also be seen that the base flow is influenced by antecedent rainfall. Mean absolute relative error was calculated 0.076 for validation data set in the combination No. 5. Coefficient of determination (R2) is another criterion for testing and it is seen from Table 3 that it's values are close to unity. We should compare our study to similar studies to determine whether our R-squared is in the right ballpark. One of the similar paper is Khan and Coulibaly (2006) that used a



Bayesian learning approach to train a multilayer feedforward network for daily river flow and reservoir inflow simulation. Their result also showed a high R-squared value. The results showed that the BN is an efficient method for modeling and combining the ensemble flood forecasts prediction. The proposed BN approach in this study predicts flood peak flow. Since the Kan River in the studied reach is a mountainous river without any flood plain storage, the peak discharge is almost not

reduced by flood routing along the river, and so we can use the peak flood instead of routing the flood hydrograph. However, in our study, we consider peak flow as the variable of interest. In other fields of application, flow volume or time to peak might be of interest.

Moreover, Bayesian cluster analysis could also provide probabilistic results for flood early warning, but since the data sample is relatively small in this study, cluster analysis cannot be achieved. This method can be also tested in basins with

sufficient historical hydrological data in the future works.

Performance of the BN model is compared with the results obtained from an ANN model as a widely used model as a benchmark. The comparison is conducted using the same data set for training and validation. These results are presented in section 3.3.

### 3.3. Artificial Neural Network Verification

The first step in developing an ANN model is to determine the input and output variables. The output from the model is the flood peak discharge magnitude, and the input variables have been selected the same of the best combination of predictor variables in BN that has been used in this study (Table 3, combination No. 5). The feed forward error back propagation algorithm has been employed as the training algorithm in this study. The difficult task in working with ANN contains selecting parameters such as the number of hidden nodes. There is no established algorithm until now to determine how

many hidden nodes are required to approximate any given function. Here, we use the common trial and error method to choose the number of hidden nodes, which are varied from 2 to 6 according to the previous studies (
bib et al., 2015). Error index is usually used to select the best performance of network model compared to observed data. The accuracy of the model for different numbers of nodes in the hidden layer is presented in Table 4. It was found that four hidden nodes give the best results. Mean absolute relative error (MARE) was calculated as 0.39 for the validation data set

while this index was calculated 0.076 in BN. The comparison shows that BN offers better accuracy. Although our data set was relatively small, the result of BN model was accurate enough. Therefore, it seems that BN is less sensitive to small data set size, so it is more suited for rare events such as floods, where the available data are limited due to the high return period of such events.

### 4. Conclusions

This study proposed a probabilistic model to address the uncertainties of flood forecasts using the Bayesian networks (BN) and to estimate the flood peak in an ensemble flood forecasting. This is the first attempt to use BN in ensemble flood forecasting. The Weather Research and Forecasting (WRF) model was used to simulate some historic precipitation rainfall



events using five various cumulus parameterization schemes. The results showed that there is no considerable decrease in accuracy by deleting the Multi-scale Kain-Fritsch scheme, thus it can be concluded that is the least accurate cumulus scheme. It also was found that Kain-Fritsch is the most efficient cumulus parameterization scheme. Atmospheric ensemble forecasts were coupled to the Bayesian Network to estimate the flood magnitude in an ensemble forecasting. Results of the

BN are compared with the results obtained from an artificial neural network as a widely used model to show the performance of BN. The comparison is conducted using the same data set for validation and training. The results showed that the BN is an efficient method for flood forecasting based on ensemble rainfall forecasts and offers better accuracy than ANN. We showed that BN is less sensitive to small data set size in comparison with other models, thus it is more suited for rare events such as floods. The results of this study indicate that BN might be a suitable tool for a fast computation of peak flow and flood

warnings from numerical ensemble weather predictions. Therefore, the result of this may tested for increasing flood forecasting lead-time using ensemble prediction of rainfalls.

**Acknowledgment**

Part of this study was carried out in Leibniz Universität Hannover during PhD sabbatical by first author, and corresponding author advised her during this period. We appreciate Leibniz Universität Hannover for supporting this study.

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





**Tables and Figures**

Table 1. Common Cumulus Parameterization Schemes

| Model | Reference | Software used |
|---|---|---|
| Kain-Fritsch (KF) | (Kain and Fritsch, 1990) | WRF version 3.8 |
| Betts-Miller-Janjic (BMJ) | (Janjić, 1994) | WRF version 3.8 |
| Grell 3D ensemble (GR3D) | (Grell, 1993) | WRF version 3.8 |
| Multi-scale Kain-Fritsch (MSKF) | (Zheng et al., 2016) | WRF version 3.8 |
| Grell-Devenyi ensemble (GDE) | (Grell and Dévényi, 2002) | WRF version 3.8 |

5                                        Table2. Precipitation and streamflow data

| event | Observed cumulative precipitation (mm) | Observed peak flow(m3/s) |
|---|---|---|
| 27.03.2007 | 25.3 | 24.2 |
| 27.04.2007 | 33.5 | 57.1 |
| 07.12.2007 | 32.3 | 12.7 |
| 03.11.2008 | 37.3 | 20.9 |
| 30.04.2009 | 29 | 34.4 |
| 04.02.2010 | 68.1 | 11.6 |
| 08.04.2010 | 48.8 | 34.1 |
| 13.03.2011 | 32.6 | 20.9 |
| 05.04.2011 | 55.5 | 24.5 |
| 29.08.2011 | 56.4 | 26.4 |
| 28.10.2011 | 55.9 | 55.1 |
| 20.11.2011 | 48 | 44.7 |
| 14.04.2012 | 67.7 | 67.2 |
| 13.11.2012 | 78.9 | 25 |




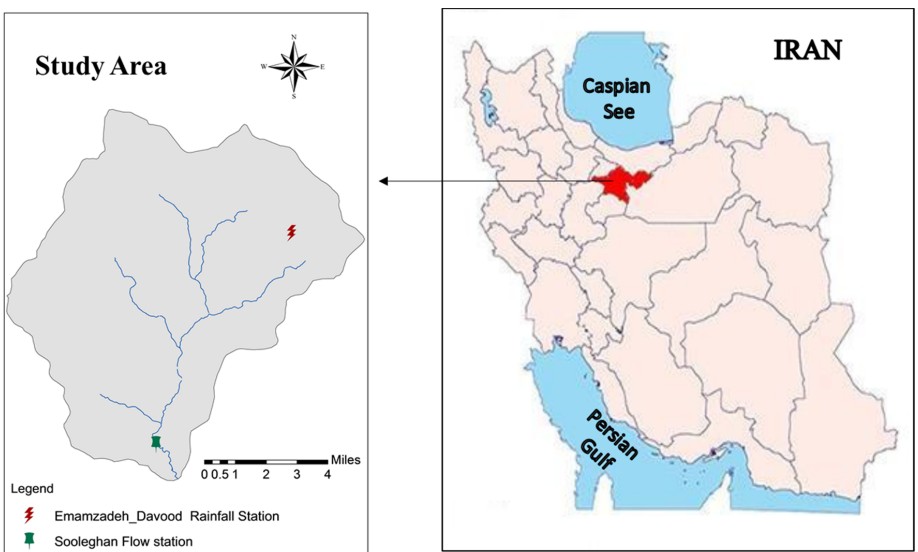

Figure 1. Location of study area, rainfall and flow stations.



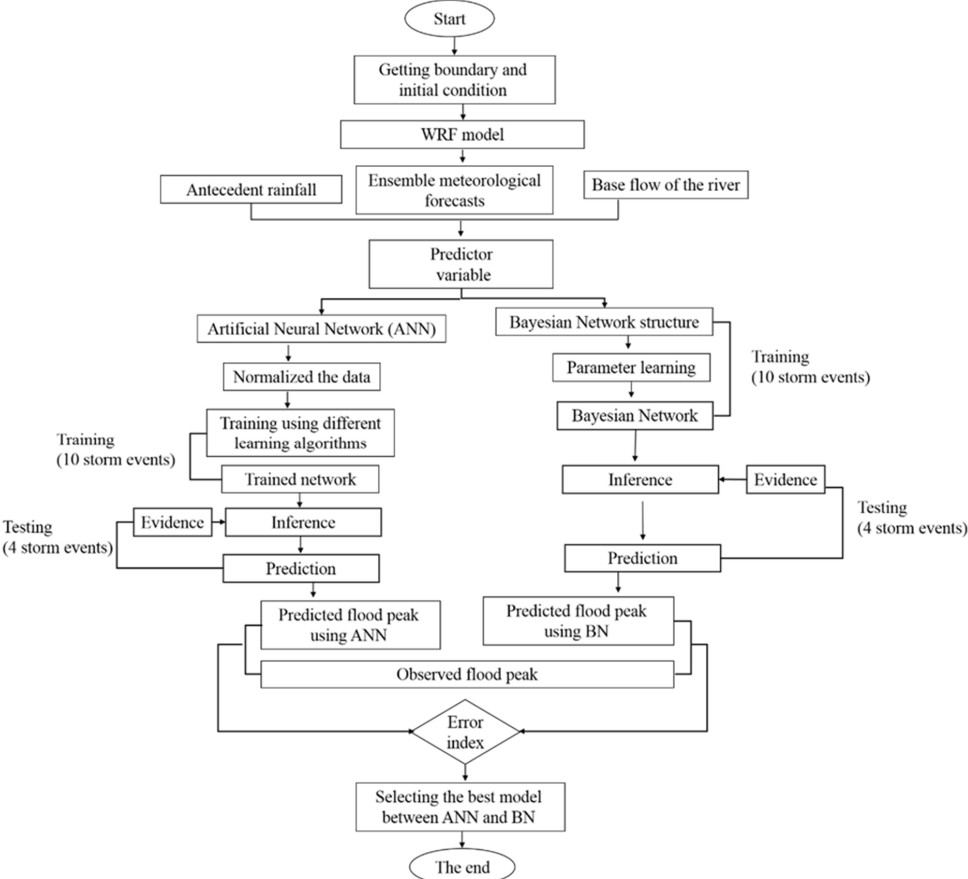

Figure 2. Flow chart of the flood forecast approach in this research



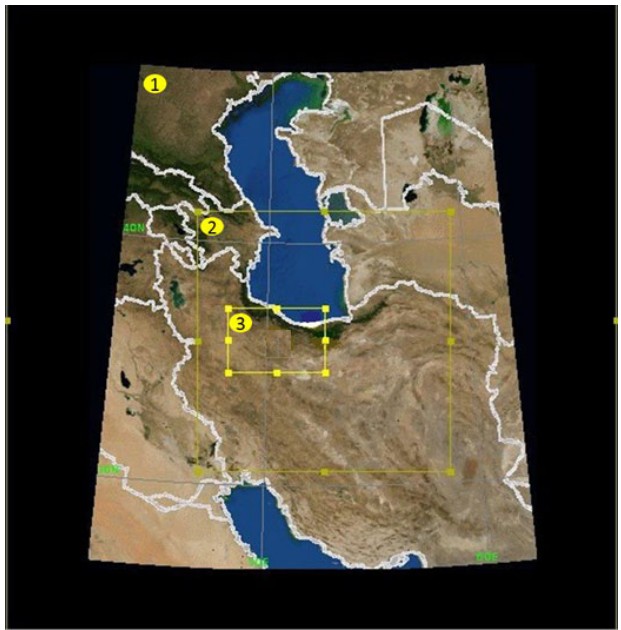

Figure 3. WRF domain setup using an inter-active nested domain inside the parent domain.

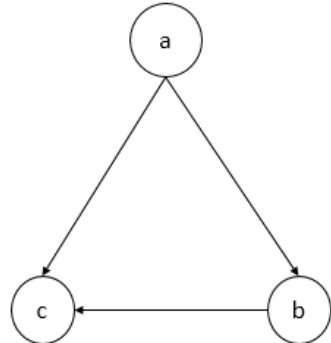

Figure 4. An example of graphical Bayesian network



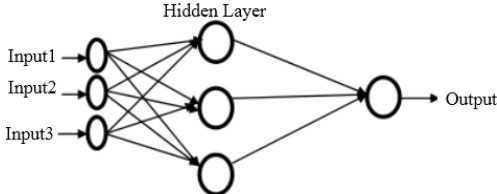

Figure 5. Typical ANN architecture

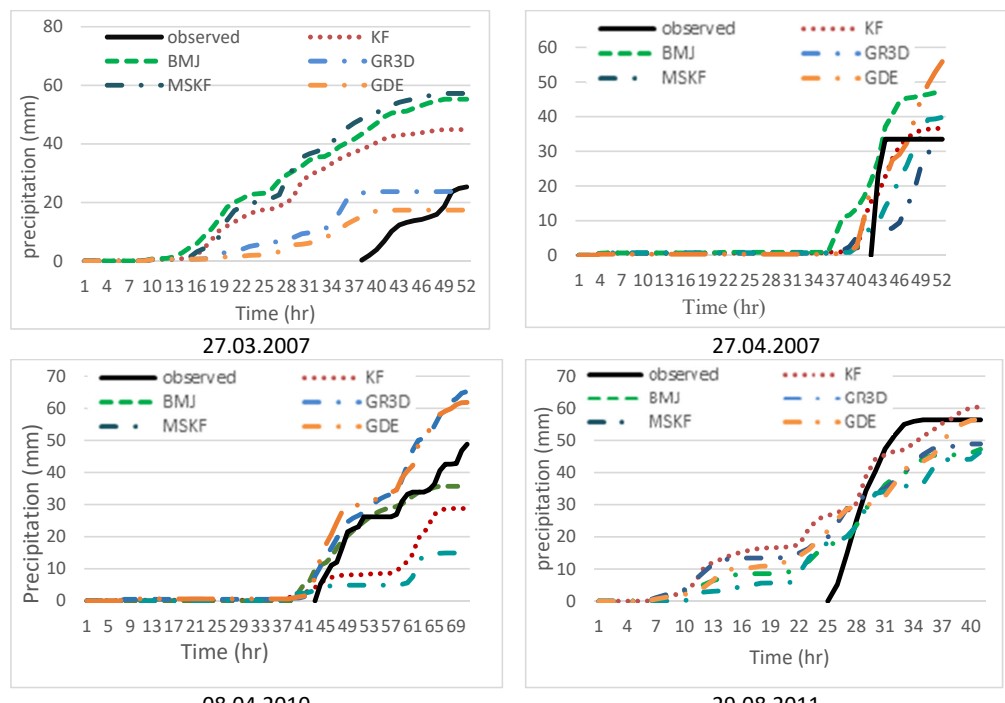

Figure 6. The ensemble forecasted precipitation and the observed cumulative precipitation



Table 3. Performance of Bayesian Network for different combinations of predictor variables.

| Combination No. | predictor variables | R2 | MARE |
|---|---|---|---|
| 1 | Maximum hourly rainfall | 0.99 | 0.16 |
| 2 | Accumulated rainfall | 0.74 | 1.06 |
| 3 | Maximum hourly rainfall, Base flow of the river | 0.99 | 0.18 |
| 4 | Maximum hourly rainfall, Antecedent rainfall | 0.99 | 0.12 |
| 5 | Maximum hourly rainfall, Base flow of the river, Antecedent rainfall | 0.99 | 0.076 |
| 6 | Maximum hourly rainfall (deleting  KF) , Base flow of the river, Antecedent soil moisture | 0.58 | 0.46 |
| 7 | Maximum hourly rainfall (deleting BMJ) , Base flow of the river, Antecedent rainfall | 0.99 | 0.23 |
| 8 | Maximum hourly rainfall (deleting GR3D) , Base flow of the river, Antecedent rainfall | 0.99 | 0.15 |
| 9 | Maximum hourly rainfall (deleting MSKF) , Base flow of the river, Antecedent rainfall | 0.99 | 0.087 |
| 10 | Maximum hourly rainfall (deleting GDE) , Base flow of the river, Antecedent rainfall | 0.99 | 0.10 |

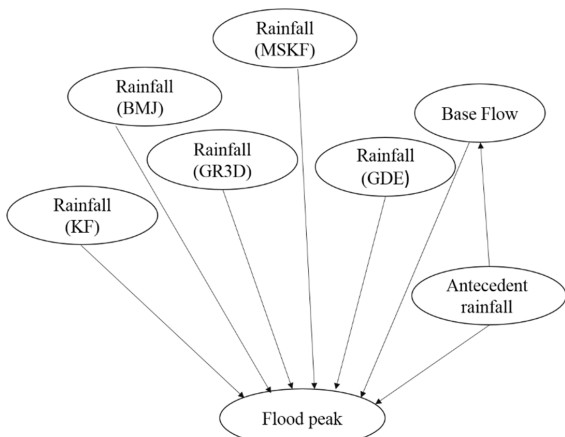

Figure 7. The proposed structure of the Bayesian network for ensemble flood forecasting



Table 4. MARE and R2 of Artificial Neural Network in Verification Phase

| Number of nodes in hidden layer | MARE | R2 |
|:---:|:---:|:---:|
| 2 | 1.14 | 0.44 |
| 3 | 0.74 | 0.92 |
| 4 | 0.39 | 0.77 |
| 5 | 0.51 | 0.93 |
| 6 | 1.23 | 0.12 |