# Peer review of "Bayesian Network Model for Flood Forecasting Based on Atmospheric Ensemble Forecasts"

_Natural Hazards and Earth System Sciences, 2019_

## Referee Comment (RC1) · Anonymous Referee #1 · 2 Jul 2019

The manuscript explores the use of Bayesian Network for flood forecasting using ensemble weather forecasts as input. The performance of the BN is compared against Artificial neural network – ANN and the authors conclude that BN outperform ANN. While exploring new methodologies and approaches for improving flood forecasting is always welcome, I have doubts about the reliability of those approaches compared to traditional methodologies, specifically the use of post-processing ensemble weather forecast as input of a distributed or lumped hydrological model. Usually, hydrological models are calibrated and validated for a long enough time-period, which ensure that they capture a wide range of hydrological conditions, including episodic floods. In this case, on the other hand, they were used 14 flood events to train and verified a BN and an ANN. Considering the large number of parameters those approaches include,

a good performance and accuracy is expected. Under such circumstances, however, the risk of generating overparameterized models is significant. Although I recognize that flood events are statistically rare, it is important to demonstrate that the BN is able to capture a larger number of floods events. Perhaps a suggestion to overcome such limitations is to analyze a longer period of time to incorporate a larger number of flood events, and using observed rainfall instead of ensemble weather forecast (which are difficult to implement) to test whether the BN performs adequately. Finally, I don't know why the forecasting were limited to 24 h. It is well known that atmospheric models have acceptable skill scores for up to 4-5 days. Increasing the lead time will provide an opportunity for testing the use of BN for a larger number of cases.

---

## Referee Comment (RC2) · Anonymous Referee #2 · 30 Jul 2019

The manuscript proposes the utilization of a Bayesian Network Model for flood fore-casting. The model utilizes the outcome from atmospheric ensemble forecasts and the results are compared against an Artificial Neural Network methodology. Consid-ering the results, the better performance of the first approach is highlighted. There are some major drawbacks that should be mentioned. Beginning with the part of the Atmospheric modeling and the simulations performed for the analysis, it is not quite clear but the fine domain used in the model configuration might be too close to the eastern parts of the basin. The fine domain should be depicted at the same figure with the study area (Figure 1). A more important issue is the lack of important information regarding the simulations such as the spin up time, the length of each simulation etc. A table including these characteristics for the 14 cases would be useful. Apart from that,

the whole analysis is performed in order to optimize the flood forecasting skill based on atmospheric modeling and post processing. However the model is not running in a forecast mode. The initial conditions are the FNL (page 9) data from NCEP, which are a post process product. The fact that the simulations are in a hindcast mode is something that should be mentioned clearly within the manuscript. Towards this direction it would be quite interesting to also employ a model running with the NCEP forecast analysis. Additionally, it would be interesting to run the model with different forecasting horizons in order to test its performance in periods with higher uncertainties. Finally the creation of an ensemble with the implementation of different cumulus schemes is something needed to be supported better, especially considering that some of them are already known to perform better than the rest. Considering the hydrological analysis, models in general are tested for larger periods, employing a range of hydrological events with different characteristics. In the proposed manuscript only 14 flood cases were implemented resulting in good results. However there is the danger of bias by taking into consideration such a small number of test cases. In any case, the meteorological characteristics for these cases should be analyzed and the cases should be divided into categories. Finally, despite the fact that such events are rare, maybe the authors should also consider taking into account smaller-impact events or maybe employing as initial conditions a dataset covering larger periods.

---

## Author Comment (AC1) · 21 Aug 2019

We would like to express our sincere gratitude for the insightful comments. Please see below the responses to the comments, on behalf of the authors.

Comment 1:

"I have doubts about the reliability of those approaches compared to traditional methodologies, specifically the use of post-processing ensemble weather forecast as input of a distributed or lumped hydrological model. Usually, hydrological models are calibrated and validated for a long enough time-period, which ensure that they capture a wide range of hydrological conditions, including episodic floods. In this case, on the other hand, they were used 14 flood events to train and verified a BN and an ANN. Considering the large number of parameters those approaches include, a good performance and accuracy is expected. Under such circumstances, however, the risk of generating over parameterized models is significant. Although I recognize that flood events are statistically rare, it is important to demonstrate that the BN is able to capture a larger number of floods events."

Response 1:

The data sample is relatively small due to the following reasons: 1) NCEP (GFS - FNL) data are not available for some historical storms. 2) During the above-mentioned period, a small number of actual flood events occurred in the study area, since the basin is located in a semi-arid region. Considering the relatively small sample size, we proposed using the BN that is less sensitive to small data set size in comparison with ANN. We are aware that using a BN instead of a hydrological model does not remove the need for data, and we agree that data about flood events are scarce by nature. However, the number of parameters of a BN is not that high compared to distributed hydrological models. Our study is a proof of concept at the current stage that flood warnings can be done by evaluating hydrological pre-conditions and mete- orological ensembles by a trained BN instead of a hydrological model. We do not yet promise that the method works in general, and further work must be done, thus we recommended future tests in the conclusions. We discussed our results accordingly. However, with the limitations described, the validation of the BN is given by the proof of better performance than the ANN. A very useful advantage of BN is that there are no minimum sample data sizes needed to perform the analysis, and BN take into account the complete data set (Myllymaki et al., 2002). In addition, Kontkanen et al. (1997) demonstrate that BN can show good accuracy of prediction even with rather small data set and Zhang and Bivens (2007) showed that BN is less sensitive to small data set size in comparison with ANN. The above paragraph will be added to paper.

Comment 2:

"Perhaps a suggestion to overcome such limitations is to analyze a longer period of time to incorporate a larger number of flood events, and using observed rainfall instead of ensemble weather forecast (which are difficult to implement) to test whether the BN performs adequately."

Response 2:

The purpose of this study is to develop a flood warning based on Atmospheric Ensemble Forecasts. BN model's input are Atmospheric Ensemble Forecasts and in case of using the observed rainfall, we have only a deterministic forecasting not the ensemble forecasting and that is why we didn't use the observed rainfall in our study. A BN trained against observation would not be comparable with the training against forecast ensembles. In the outlook of the article, we propose other steps to increase the confidence in the BN by increasing the lead time in large watersheds, using different cumulus schemes, etc.

Comment 3:

"It is well known that atmospheric models have acceptable skill scores for up to 4-5 days. Increasing the lead time will provide an opportunity for testing the use of BN for a larger number of cases."

Response 3:

Increasing the lead time will provide new cases but in this case we have two different sources of error: one is the different lead time (the accuracy of the numerical weather prediction would not be comparable to a single day lead time) and another source is the BN model, so we cannot realize the source of the error. In other words, we cannot determine that the forecasting error is because of the high lead time or the proposed BN model. Also, our study is conducted in a small basin, where a lead time of one day is considered sufficient and adequate. Longer lead times are more important for large watersheds, but there is a different ratio between meteorological and hydrological

effects. Thus, our method is designed for, and limited to, smaller headwater basins with short lead time. We will make this clearer in the final manuscript.

Once again, we wish to express our highest appreciation to the reviewers for their comments. We provided a first study of a new method in flood warning, which still has some limitations and much further work is required to get more insights and knowledge about general applicability. We hope the manuscript will suit the Journal Natural Hazards and Earth System Sciences and we are happy to provide a revised manuscript. We thank you for your continued interest in our research.

Yours sincerely

The Authors

References:

Li, J., Chen, Y., Wang, H., Qin, J., Li, J., & Chiao, S. (2017). Extending flood forecasting lead time in a large watershed by coupling WRF QPF with a distributed hydrological model. Hydrology and Earth System Sciences, 21(2), 1279-1294. Myllymaki, P., Silander, T., Tirri, H., and Uronen, P. (2002). B-Course: a web-based tool for Bayesian and causal data analysis. Int. J. Artif. Intell. Tools 11 (3), 369–387, doi: 10.1142/s0218213002000940. Kontkanen, P., Myllymaki, P., Silander, T., and Tirri, H. (1997). Comparing predictive inference methods for discrete domains. In: Proceedings of the sixth International Workshop on Artificial Intelligence and Statistics, Ft. Lauderdale, USA, 311–318. Zhang, R. and Bivens, A.J. (2007). Comparing the use of bayesian networks and neural networks in response time modeling for service-oriented systems. In: Proceedings of the 2007 workshop on Service-oriented computing performance: aspects, issues, and approaches (pp. 67-74). ACM.
* * *

---

## Author Comment (AC2) · 21 Aug 2019

We would like to express our sincere gratitude for the insightful comments. Please see below the responses to the comments, on behalf of the authors.

Comment 1:

"Beginning with the part of the Atmospheric modeling and the simulations performed for the analysis, it is not quite clear but the fine domain used in the model configuration might be too close to the eastern parts of the basin. The fine domain should be depicted at the same figure with the study area (Figure 1)."

Response 1:

[Figure]

Thank you very much for catching this confusing issue, which we will clarify in the next version. We should explain that the red region in Figure 1 is all of the Tehran province. Our case study is a small basin in the north western part of this province, so the model configuration is not close to eastern part and we will correct the Figure 1 in new manuscript.

Comment 2:

"A more important issue is the lack of important information regarding the simulations such as the spin up time, the length of each simulation etc. A table including these characteristics for the 14 cases would be useful."

Response 2:

We will add a table in the next version, see supplement file.

Comment 3:

"The initial conditions are the FNL (page 9) data from NCEP, which are a post process product. The fact that the simulations are in a hindcast mode is something that should be mentioned clearly within the manuscript. Towards this direction it would be quite interesting to also employ a model running with the NCEP forecast analysis. Additionally, it would be interesting to run the model with different forecasting horizons in order to test its performance in periods with higher uncertainties. Finally the creation of an ensemble with the implementation of different cumulus schemes is something needed to be supported better, especially considering that some of them are already known to perform better than the rest."

Response 3:

Thank you for this suggestion. We will mention the use of hindcast mode in the next version. Regarding the use of different forecasting horizons, please see our response to reviewer 1. It is an interesting aspect, but we focused on the short term as this is the recommended lead time for the size of catchment under consideration. Long lead time

flood forecasting is very important for large watershed flood mitigation as it provides more time for flood warning and emergency responses (Li et al, 2017). Further work may deal with the transferability of BN to longer lead times and other catchments, and investigate the need for re-training of the BN based on the different characteristics of the meteorological uncertainty for the different lead times, and based on the different catchment characteristics and this can be recommended for future studied in the conclusion. We have used five various cumulus parameterization schemes. Running the model with more cumulus schemes would have been interesting to explore this aspect. However, in the case of our study, it seems out of scope because the purpose of our study is to propose the Bayesian Network (BN) model to estimate flood peak in case of small data size like flood forecasting. In other words, we focused on the hydrological forecasting aspects in our paper. However, we agree that using more cumulus schemes might improve the prediction so, following the reviewer suggestion, we will propose using different cumulus schemes for future work to explore the uncertainties of the meteorological forecast better.

Comment 4:

"In the proposed manuscript only 14 flood cases were implemented resulting in good results. However there is the danger of bias by taking into consideration such a small number of test cases. In any case, the meteorological characteristics for these cases should be analyzed and the cases should be divided into categories. Finally, despite the fact that such events are rare, maybe the authors should also consider taking into account smaller-impact events or maybe employing as initial conditions a dataset covering larger periods."

Response 4:

We agree with the limitation of the small sample size. Using categories of events can be useful, but would even further reduce the sample size for the different categories. With the small number of events in total, we did not attempt to divide the dataset further

and train the BN for the different categories. As it is a semi-arid catchment, we assume that rainfall characteristics is implicitly regarded by the incorporation of our input variables for the BN, which we have checked and documented. In particular, hydrological initial conditions showed to be relevant. A very useful advantage of BN is that there are no minimum sample data sizes needed to perform the analysis, and BN take into account the complete data set (Myllymaki et al., 2002). Also, Kontkanen et al. (1997) demonstrate that BN can show good accuracy of prediction even with rather small data set. Furthermore, Zhang and Bivens (2007) showed that BN is less sensitive to small data set size in comparison with ANN. It is a good idea to include smaller events in order to have more data, but these events would not have relevance for flood warning, and their characteristics is most probably much different, so there would maybe be a trade-off in training the BN for the large and the small events at the same time. We will extend our outlook regarding that aspect.

Once again, we wish to express our highest appreciation to the reviewers for their comments. We provided a first study of a new method in flood warning, which still has some limitations and much further work is required to get more insights and knowledge about general applicability. We hope the manuscript will suit the Journal Natural Hazards and Earth System Sciences and we are happy to provide a revised manuscript. We thank you for your continued interest in our research.

Yours sincerely

The Authors

References:

Li, J., Chen, Y., Wang, H., Qin, J., Li, J., & Chiao, S. (2017). Extending flood forecasting lead time in a large watershed by coupling WRF QPF with a distributed hydrological model. Hydrology and Earth System Sciences, 21(2), 1279-1294. Myllymaki, P., Silander, T., Tirri, H., and Uronen, P. (2002). B-Course: a web-based tool for Bayesian and causal data analysis. Int. J. Artif. Intell. Tools 11 (3), 369–387,

doi: 10.1142/s0218213002000940. Kontkanen, P., Myllymaki, P., Silander, T., and Tirri, H. (1997). Comparing predictive inference methods for discrete domains. In: Proceedings of the sixth International Workshop on Artificial Intelligence and Statistics, Ft. Lauderdale, USA, 311–318. Zhang, R. and Bivens, A.J. (2007). Comparing the use of bayesian networks and neural networks in response time modeling for service-oriented systems. In: Proceedings of the 2007 workshop on Service-oriented computing performance: aspects, issues, and approaches (pp. 67-74). ACM.

Please also note the supplement to this comment:
https://www.nat-hazards-earth-syst-sci-discuss.net/nhess-2019-44/nhess-2019-44-AC2-supplement.pdf

**Supplement:**

Table2. Precipitation and streamflow data

| event | Observed cumulative precipitation (mm) | Observed peak flow(m3/s) | Duration (hr) |
|---|---|---|---|
| 27.03.2007 | 25.3 | 24.2 | 15 |
| 27.04.2007 | 33.5 | 57.1 | 2 |
| 07.12.2007 | 32.3 | 12.7 | 17 |
| 03.11.2008 | 37.3 | 20.9 | 17 |
| 30.04.2009 | 29 | 34.4 | 7 |
| 04.02.2010 | 68.1 | 11.6 | 11 |
| 08.04.2010 | 48.8 | 34.1 | 29 |
| 13.03.2011 | 32.6 | 20.9 | 14 |
| 05.04.2011 | 55.5 | 24.5 | 25 |
| 29.08.2011 | 56.4 | 26.4 | 11 |
| 28.10.2011 | 55.9 | 55.1 | 23 |
| 20.11.2011 | 48 | 44.7 | 31 |
| 14.04.2012 | 67.7 | 67.2 | 15 |
| 13.11.2012 | 78.9 | 25 | 41 |

---

## Author Response (AR1)

**Article: Bayesian Network Model for Flood Forecasting Based on Atmospheric Ensemble Forecasts**

By: Leila Goodarzi et al.

**Response to the comments of the editor and reviewers (text with marked changes attached):**

Dear Reviewer and Associate Editor,

We would like to express our sincere gratitude for the insightful comments. Please see below our response to the comments. We have attached the new manuscript, where we have marked major changes in yellow color. Furthermore, we have made many formal corrections, mostly in the citations/list of references, which we have thoroughly checked for consistency.

**Editor:**

Many thanks for your response to the referees.

I think, that most of the referee comments have been addressed. But we need to check the new version of the manuscript.

Please in this new version include some comments about the sample size and the limitations of the method.

Thanks for your work."

**Author response:**

Thank you very much for appreciating our work. We have now incorporated the proposed changes of the text. In particular, we have extended the justification of using BN with a small sample size, and we discussed more extended about that aspect, limitations and future work. Our responses are the same as uploaded in the interactive discussion. Here, they are provided again together with the updated manuscript, where the major changes are marked.

**Reviewer 1:**

1)        I have doubts about the reliability of those approaches compared to traditional methodologies, specifically the use of post-processing ensemble weather forecast as input of a distributed or lumped hydrological model. Usually, hydrological models are calibrated and validated for a long enough time-period, which ensure that they capture a wide range of hydrological conditions, including episodic floods. In this case, on the other hand, they were used 14 flood events to train and verified a BN and an ANN. Considering the large number of parameters those approaches include, a good performance and accuracy is expected. Under such circumstances, however, the risk of generating over parameterized models is significant. Although I recognize that flood events are statistically rare, it is important to demonstrate that the BN is able to capture a larger number of floods events.

**Author response:**

The data sample is relatively small due to the following reasons:

1) NCEP (GFS - FNL) data are not available for some historical storms.

2) During the above-mentioned period, a small number of actual flood events occurred in the study area, since the basin is located in a semi-arid region.

Considering the relatively small sample size, we proposed using the BN that is less sensitive to small data set size in comparison with ANN. We are aware that using a BN instead of a hydrological model does not remove the need for data, and we agree that data about flood events are scarce by nature. However, the number of parameters of a BN is not that high compared to distributed hydrological models. Our study is a proof of concept at the current stage that flood warnings can be done by evaluating hydrological pre-conditions and meteorological ensembles by a trained BN instead of a hydrological model. We do not yet promise that the method works in general, and further work must be done, thus we recommended future tests in the conclusions. We discussed our results accordingly. However, with the limitations described, the validation of the BN is given by the proof of better performance than the ANN.

A very useful advantage of BN is that there are no minimum sample data sizes needed to perform the analysis, and BN take into account the complete data set (Myllymaki et al., 2002). In addition, Kontkanen et al. (1997) demonstrate that BN can show good accuracy of prediction even with rather small data set and Zhang and Bivens (2007) showed that BN is less sensitive to small data set size in comparison with ANN.

The above paragraph will be added to paper.

2)     Perhaps a suggestion to overcome such limitations is to analyze a longer period of time to incorporate a larger number of flood events, and using observed rainfall instead of ensemble weather forecast (which are difficult to implement) to test whether the BN performs adequately.

**Author response:** The purpose of this study is to develop a flood warning based on Atmospheric Ensemble Forecasts. BN model's input are Atmospheric Ensemble Forecasts and in case of using the observed rainfall, we have only a deterministic forecasting not the ensemble forecasting and that is why we didn't use the observed rainfall in our study. A BN trained against observation would not be comparable with the training against forecast ensembles. In the outlook of the article, we propose other steps to increase the confidence in the BN by increasing the lead time in large watersheds, using different cumulus schemes, etc.

3)     It is well known that atmospheric models have acceptable skill scores for up to 4-5 days. Increasing the lead time will provide an opportunity for testing the use of BN for a larger number of cases.

**Author response:** Increasing the lead time will provide new cases but in this case we have two different sources of error: one is the different lead time (the accuracy of the numerical weather prediction would not be comparable to a single day lead time) and another source is the BN model, so we cannot realize the source of the error. In other words, we cannot determine that the forecasting error is because of the high lead time or the proposed BN model. Also, our study is conducted in a small basin, where a lead time of one day is considered sufficient and adequate. Longer lead times are more important for large watersheds, but there is a different ratio between meteorological and hydrological effects. Thus, our method is designed for, and limited to, smaller headwater basins with short lead time. We will make this clearer in the final manuscript.

1)        Beginning with the part of the Atmospheric modeling and the simulations performed for the analysis, it is not quite clear but the fine domain used in the model configuration might be too close to the eastern parts of the basin. The fine domain should be depicted at the same figure with the study area (Figure 1).

5   **Response:** Thank you so much for catching this confusing issue, which we will clarify in the next version. We should explain that the red region in Figure 1 is all of the Tehran province. Our case study is a small basin in the north western part of this province, so the model configuration is not close to eastern part and we will correct the Figure 1 in new manuscript.

2) A more important issue is the lack of important information regarding the simulations such as the spin up time, the length of each simulation etc. A table including these characteristics for the 14 cases would be useful.

10   **Response:** we will add a table like the following table in the next version.

Table2. Precipitation and streamflow data

| event | Observed cumulative precipitation (mm) | Observed peak flow(m3/s) | Duration (hr) |
|---|---|---|---|
| 27.03.2007 | 25.3 | 24.2 | 15 |
| 27.04.2007 | 33.5 | 57.1 | 2 |
| 07.12.2007 | 32.3 | 12.7 | 17 |
| 03.11.2008 | 37.3 | 20.9 | 17 |
| 30.04.2009 | 29 | 34.4 | 7 |
| 04.02.2010 | 68.1 | 11.6 | 11 |
| 08.04.2010 | 48.8 | 34.1 | 29 |
| 13.03.2011 | 32.6 | 20.9 | 14 |
| 05.04.2011 | 55.5 | 24.5 | 25 |
| 29.08.2011 | 56.4 | 26.4 | 11 |
| 28.10.2011 | 55.9 | 55.1 | 23 |
| 20.11.2011 | 48 | 44.7 | 31 |
| 14.04.2012 | 67.7 | 67.2 | 15 |
| 13.11.2012 | 78.9 | 25 | 41 |

3) The initial conditions are the FNL (page 9) data from NCEP, which are a post process product. The fact that the simulations are in a hindcast mode is something that should be mentioned clearly within the manuscript. Towards this direction it would

15   be quite interesting to also employ a model running with the NCEP forecast analysis. Additionally, it would be interesting to run the model with different forecasting horizons in order to test its performance in periods with higher uncertainties. Finally the creation of an ensemble with the implementation of different cumulus schemes is something needed to be supported better, especially considering that some of them are already known to perform better than the rest.

**Response:** Thank you for this suggestion. We will mention the use of hindcast mode in the next version. Regarding the use of different forecasting horizons, please see our response to reviewer 1. It is an interesting aspect, but we focused on the short term as this is the recommended lead time for the size of catchment under consideration. Long lead time flood forecasting is very important for large watershed flood mitigation as it provides more time for flood warning and emergency responses (Li et al, 2017). Further work may deal with the transferability of BN to longer lead times and other catchments, and investigate the need for re-training of the BN based on the different characteristics of the meteorological uncertainty for the different lead times, and based on the different catchment characteristics and this can be recommended for future studied in the conclusion. We have used five various cumulus parameterization schemes. Running the model with more cumulus schemes would have been interesting to explore this aspect. However, in the case of our study, it seems out of scope because the purpose of our study is to propose the Bayesian Network (BN) model to estimate flood peak in case of small data size like flood forecasting. In other words, we focused on the hydrological forecasting aspects in our paper. However, we agree that using more cumulus schemes might improve the prediction so, following the reviewer suggestion, we will propose using different cumulus schemes for future work to explore the uncertainties of the meteorological forecast better

4) In the proposed manuscript only 14 flood cases were implemented resulting in good results. However there is the danger of bias by taking into consideration such a small number of test cases. In any case, the meteorological characteristics for these cases should be analyzed and the cases should be divided into categories. Finally, despite the fact that such events are rare, maybe the authors should also consider taking into account smaller-impact events or maybe employing as initial conditions a dataset covering larger periods.

**Author response:** We agree with the limitation of the small sample size. Using categories of events can be useful, but would even further reduce the sample size for the different categories. With the small number of events in total, we did not attempt to divide the dataset further and train the BN for the different categories. As it is a semi-arid catchment, we assume that rainfall characteristics is implicitly regarded by the incorporation of our input variables for the BN, which we have checked and documented. In particular, hydrological initial conditions showed to be relevant. A very useful advantage of BN is that there are no minimum sample data sizes needed to perform the analysis, and BN take into account the complete data set (Myllymaki et al., 2002). Also, Kontkanen et al. (1997) demonstrate that BN can show good accuracy of prediction even with rather small data set. Furthermore, Zhang and Bivens (2007) showed that BN is less sensitive to small data set size in comparison with ANN. It is a good idea to include smaller events in order to have more data, but these events would not have relevance for flood warning, and their characteristics is most probably much different, so there would maybe be a trade-off in training the BN for the large and the small events at the same time. We will extend our outlook regarding that aspect.

*Once again, we wish to express our* highest *appreciation* to the *reviewers* for *their comments*. We provided a first study of a new method in flood warning, which still has some limitations and much further work is required to get more insights and knowledge about general applicability. We hope the manuscript will suit the Journal Natural Hazards and Earth System Sciences and we are happy to provide a revised manuscript. We thank you for your continued interest in our research.

Yours sincerely,

the authors

5    (References cited in our response can be found in the updated manuscript)

[revised manuscript text omitted]

---

## Referee Report (RR1)

The revised manuscript regarding the utilization of a Bayesian Network Model for flood forecasting is improved. However, the second comment was partially covered. The table that was attached within "the replies to reviewers" part is not the same as the one imported within the manuscript. The Duration of the storms and the duration of the runs are not evident.

---

## Author Response (AR2)

Dear Editor,

thank you very much for handling our manuscript. Sorry for the mistake with the missing column in table 2. We have exchanged the table as announced in our response. We have also modified the sentence referring to the table. We saw one error in the numbering of the tables: we have changed the number of the last table to "5", and also changed the reference in the text.

Figures have been technically optimized for publishing.

Best regards

Jörg Dietrich (for the authors)